# Age-differentiated incentives for adaptive behavior during epidemics produce oscillatory and chaotic dynamics

**Ronan F. Arthur[1], May Levin[2], Alexandre Labrogere[3], Marcus W. Feldman[2]***

**1** School of Medicine, Stanford University, Stanford, California, United States of America, **2** Department of Biology, Stanford University, Stanford, California, United States of America, **3** Department of Management Science & Engineering, Stanford University, Stanford, California, United States of America

* mfeldman@stanford.edu

**Data Availability Statement:** All relevant data are within the manuscript. Code for the modeling can be found at the following website: https://github.com/levinmay/adaptiveBehavior.

## Abstract

Heterogeneity in contact patterns, mortality rates, and transmissibility among and between different age classes can have significant effects on epidemic outcomes. Adaptive behavior in response to the spread of an infectious pathogen may give rise to complex epidemiological dynamics. Here we model an infectious disease in which adaptive behavior incentives, and mortality rates, can vary between two and three age classes. The model indicates that age-dependent variability in infection aversion can produce more complex epidemic dynamics at lower levels of pathogen transmissibility and that those at less risk of infection can still drive complexity in the dynamics of those at higher risk of infection. Policymakers should consider the interdependence of such heterogeneous groups when making decisions.

## Author summary

Behavior during epidemics is driven by incentives, which may vary among different age classes. Here we include age-differentiated incentives in an adaptive behavioral epidemic model with two and three age classes. Our results indicate that this heterogeneity can drive the system into oscillations, chaos, and collapse. Those with less incentives to adopt protective behavior for themselves (i.e., the youth) can still drive epidemic dynamics and associated complexity in those who have more incentives to avoid contacts during epidemics (i.e., the elderly). This heterogeneity and interdependence should be taken into account when making public health policy that affects individual contact rates during an epidemic.

## Introduction

One of the principal failings of the attempts to model and predict future trends and dynamics of infectious disease epidemics has been the lack of incorporation of human behavior into these models [1]. The social drivers of infectious disease dynamics have been relatively neglected in the academic literature compared with the vast resources and attention paid to

**Funding:** RA received support from the John Templeton Foundation, grant #61809. The funders had no role in study design, data collection and analysis, decision to publish, or preparation of the manuscript.

**Competing interests:** The authors report no competing interests.

the biomedical and, to a lesser extent, the ecological drivers of disease [2, 3]. COVID-19, however, has shone a spotlight on this discrepancy as socio-cultural factors have played an important role in the pandemic—from the political polarization of risk perception in the United States [4] to the social, cultural, and demographic factors associated with vaccine hesitancy [5]. Models should incorporate relevant social phenomena, as well as their interactions, as the study of each phenomenon in isolation may not be informative or useful. For example, adaptive age-specific preventative behaviors motivated by differentiated risk of COVID-19 mortality should be included. The resulting contact patterns and associated infection dynamics are expected to interact in their influence on epidemic trajectories.

The mortality rate of COVID-19 is highly age-specific [6] with a log-linear increase of infection fatality ratio by age among individuals over 30 [7]. This contributes to skewed risk perceptions across the population's age structure and economic activity associated with COVID-19 risk [8, 9]. Heterogeneity in contact patterns, mortality rates, and transmissibility among and between different age classes is known to have significant effects on epidemic outcomes [10, 11]. In Germany, for example, younger adults and teenagers were likely the main drivers of COVID-19 transmission dynamics during the first three pandemic waves [12]. Goldstein et al. [13] found that vaccinating the elderly first against COVID-19 would save the most lives, although the model neglected key features of transmission dynamics [14]. Further, Acemoglu and colleagues showed that optimal lockdown policies during COVID-19 are age-specific, with strict lockdown policies on the oldest group and reduction of interactions between age classes being most effective [15]. Their model, however, assumed exogenous targeted policies and did not incorporate adaptive behavior, a cornerstone of COVID-19 dynamics.

As risk of transmission of a dangerous infection increases during an epidemic, individuals and governments tend to react by mitigating that risk with adaptive behavior. Adaptive behavior has enjoyed growing attention in the disease modeling literature [16, 17], with techniques using game theory [18], fear-infection parallel contagions [19, 20], and network- and agent-based approaches [21]. Early work by Capasso et al. [22] experimented with the introduction of a negative feedback mechanism in the traditional susceptible-infected-recovered (SIR) model [23] and represented reduced contacts as a function of the number of infected. Philipson formalized the economic theory of adaptive behavior showing that rational agents following dynamic incentives may lead to oscillations around an indifference point, or equilibrium [24]. Adaptive behavior can cause oscillatory dynamics because a system with a negative feedback can continuously overshoot adjustments [25, 26].

Adaptive behavior may also lead to complex dynamics that are characteristic of deterministic, chaotic systems, especially with delays in adaptive response [27]. Empirical evidence from the COVID-19 pandemic suggests that the behavior of the epidemic has been chaotic in a majority of countries [28]. An investigation of the second derivative of infections over time during COVID-19 found that the pandemic qualitatively met Henrí Poincaré's criteria for chaos in deterministic dynamical systems: a large number of solutions, dynamic sensitivity, and numerical unpredictability [29]. Measles models show chaotic dynamics that may also characterize the observed dynamics [30, 31]. In the SEIR framework with a periodically varying contact rate that represented seasonal changes, sustained oscillations [32] and period-doubling bifurcations [33] were found, and one case with a particularly high degree of contact led to chaotic dynamics [34]. While age structure heterogeneity in susceptibility and seasonal variance in contact rates due to school attendance characterize measles modeling [35, 36], age structure heterogeneity in dynamic, adaptive contact rates have not been modeled for COVID-19.

Here, we model an infectious disease with adaptive behavior incentives of the form used in a previous model [27] and include different mortality rates for three age classes: the "young" (ages 20-49), the "middle-aged" (50-64), and the "old" (65+) (à la Acemoglu et al. [15]).

## Model specifications

We begin with a susceptible-infected-susceptible (SIS) compartmental disease model [37], which includes an adaptive contact behavior that maximizes a utility function, as in Arthur et al. [27]. With susceptible and infectious individuals, denoted by $S_t$ and $I_t$, respectively, at time $t$, and $N_t$ the population size, we have:

$$S_{t+1} = S_t - b_0 \; c_t^* S_t I_t + \gamma I_t \tag{1}$$

$$I_{t+1} = I_t + b_0 \; c_t^* S_t I_t - \gamma I_t \tag{2}$$

$$S_t + I_t = N_t, \tag{3}$$

where $b_0$ represents transmissibility given contact, $\gamma$ represents the removal rate, and $c_t^*$ represents a contact rate at time $t$ chosen to maximize utility $U(c)$ as a function of contacts:

$$U(c) = \alpha_0 - \alpha_1 (c - \hat{c})^2 - \alpha_2 \left\{ 1 - \left[ 1 - \left( \frac{I_{t-\Delta}}{N} \right) b_0 \right]^c \right\}. \tag{4}$$

Here, c represents contacts per unit time, $\hat{c}$ represents the optimal contact rate when the disease risk is non-existent, $\alpha_1$ represents the utility lost for deviating from $\hat{c}$ contacts and $\alpha_2$ represents the utility lost if infected. Here, $\Delta$ represents delayed information, such that an individual may base their perception of infection risk on prevalence during past time periods, rather than the current one. Maximizing Eq 4 with respect to c yields $c_t^*$, the contact rate chosen at each time step to maximize utility.

To incorporate age structure, we stratify the population into $k$ discrete age classes, represented by $A_1, \ldots, A_k$. The number in sub-population $A_i$ is given by $N_i$. We assume the size of each age class does not change: $N_{it} = N_i$ and let $\sum_{i=1}^{k} N_i = N$. The utility function (as in Eq 4) for each age class i is given by

$$U_i(c) = \alpha_0 - \alpha_{1i} (c - \hat{c}_i)^2 - \alpha_{2i} \left\{ 1 - \left[ 1 - \left( \frac{I_{t-\Delta}}{N} \right) b_0 \right]^c \right\}, \tag{5}$$

where $\alpha_{1i}$ represents the utility loss of reduced contacts for $A_i$, $\hat{c}_i$ represents the target contact rate for $A_i$, and $\alpha_{2i}$ represents the utility loss (i.e., aversion) to infection based on a delayed perception of population-level prevalence for $A_i$. Here it is assumed that each age class perceives its risk according to the disease prevalence of the whole population (i.e., $I_{t-\Delta} = \sum_{i=1}^{k} I_{i,t-\Delta}$), rather than just of their own group.

Interactions between and within age classes at time $t$ are defined in terms of a dynamic contact matrix $M_t$ (censu Ram & Schaposnik, 2021 [38]),

$$M_t = \begin{bmatrix} c_{1t}^* & c_{12} & \cdots & c_{1k} \\ c_{21} & c_{2t}^* & \cdots & c_{2k} \\ \vdots & \vdots & \ddots & \vdots \\ c_{k1} & c_{k2} & \cdots & c_{kt}^* \end{bmatrix}, \tag{6}$$

where $c_{it}^*$ represents the within-group contact of $A_i$, optimized at time step $t$ to maximize utility in $A_i$ (as in Eq 4), and $c_{ij}$ represents the contact between $A_i$ and $A_j$ for $i \neq j$. For simplicity, institutional behavior change is assumed to only affect the within-group contact (e.g., via school

and workplace closures and nursing home quarantines), and thus, between-group contact rates are assumed fixed and not adaptive to changing infection risks. It is assumed that $c_{ij} = c_{ji}$.

Using the contact matrix (Eq 6), the transition between susceptible and infected disease states for age class $A_i$, as in Eqs 1–3, can be expressed as

$$S_{i(t+1)} = S_{it} - b_0 c_{it}^* S_{it} I_{it} - \sum_{j=1, i \neq j}^{k} b_0 c_{ij} S_{it} I_{jt} + \gamma_i I_{it}, \tag{7}$$

$$I_{i(t+1)} = I_{it} + b_0 c_{it}^* S_{it} I_{it} + \sum_{j=1, i \neq j}^{k} b_0 c_{ij} S_{it} I_{jt} - \gamma_i I_{it}, \tag{8}$$

$$S_{it} + I_{it} = N_{it}. \tag{9}$$

We assume the following constraints:

1. The aversion to infection is greatest in the elderly and least in the young, i.e.,

$$\alpha_{2k} > \cdots > \alpha_{21}. \tag{10}$$

2. The target contact rate is greatest in the young and least in the elderly, i.e.,

$$\hat{c}_1 > \cdots > \hat{c}_k. \tag{11}$$

3. The recovery rate from the infected category to the susceptible is the same for all age classes, namely, for all $i, j$,

$$\gamma_i = \gamma_j = \gamma. \tag{12}$$

4. The number of infecteds in any age class $A_i$ may never be greater than the population size of that class or less than zero, i.e., for all i and t,

$$0 \leq I_{it} \leq N_i. \tag{13}$$

## Analytical results

### Equilibria

To understand the dynamic behavior of the number of infecteds in each age class, we first examine conditions for the existence of equilibria. If $I_t b_0$ is small relative to N, then on linearizing with respect to I in Eq 5, the optimal value of $c_i$ at time $t$ for $A_i$ is found to be

$$c_{it}^* = \hat{c}_i - \frac{\alpha_{2i} b_0 I_{t-\Delta}}{2\alpha_{1i} N}. \tag{14}$$

Define the parameter $\alpha_i$ as

$$\alpha_i = \frac{\alpha_{2i} b_0}{2\alpha_{1i}}. \tag{15}$$

We begin with a 2-age-class model, where $A_1$ represents the youth and $A_2$ represents the middle-aged and elderly, in which case, Eqs 6–13, with $k = 2$, become

$$M_t = \begin{bmatrix} c_{1t}^* & c_{12} \\ c_{12} & c_{2t}^* \end{bmatrix}, \tag{16}$$

$$I_{1(t+1)} = I_{1t} + b_0 c_{1t}^* S_{1t} I_{1t} + b_0 c_{12} S_{1t} I_{2t} - \gamma I_{1t}, \tag{17}$$

$$I_{2(t+1)} = I_{2t} + b_0 c_{2t}^* S_{2t} I_{2t} + b_0 c_{12} S_{2t} I_{1t} - \gamma I_{2t}, \tag{18}$$

with $S_{1t} + I_{1t} = N_1$, $S_{2t} + I_{2t} = N_2$, and $N = N_1 + N_2$.

For simplicity, we assume the population sizes of all groups are equal (i.e., $N_1 = N_2 = \frac{N}{2}$). Then, substituting $c_{1t}^*$ from Eq 14 and $\alpha_1$ from Eq 15, with $\Delta = 0$, we obtain

$$I_{1(t+1)} = I_{1t} - \gamma I_{1t} + b_0 \left[ \hat{c}_1 - \alpha_1 \frac{(I_{1t} + I_{2t})}{N} \right] \left( \frac{N}{2} - I_{1t} \right) I_{1t} + b_0 c_{12} I_{2t} \left( \frac{N}{2} - I_{1t} \right). \tag{19}$$

$$= f_1(I_{1t}, I_{2t}) \tag{20}$$

$$= \begin{aligned} & \frac{\alpha_1 b_0}{N} I_{1t}^3 + \left( \alpha_1 b_0 \frac{I_{2t}}{N} - b_0 \hat{c}_1 - \frac{\alpha_1 b_0}{2} \right) I_{1t}^2 + (b_0 \hat{c}_1 \frac{N}{2} - \\ & \frac{b_0 \alpha_1 I_{2t}}{2} - b_0 c_{12} I_{2t} + 1 - \gamma) I_{1t} + b_0 c_{12} I_{2t} \frac{N}{2}. \end{aligned} \tag{21}$$

By symmetry,

$$I_{2(t+1)} = \begin{aligned} & \frac{\alpha_2 b_0}{N} I_{2t}^3 + \left( \alpha_2 b_0 \frac{I_{1t}}{N} - b_0 \hat{c}_2 - \frac{\alpha_2 b_0}{2} \right) I_{2t}^2 + (b_0 \hat{c}_2 \frac{N}{2} - \\ & \frac{b_0 \alpha_2 I_{1t}}{2} - b_0 c_{12} I_{1t} + 1 - \gamma) I_{2t} + b_0 c_{12} I_{1t} \frac{N}{2}. \end{aligned} \tag{22}$$

$$= f_2(I_{1t}, I_{2t}). \tag{23}$$

A fixed point (i.e., equilibrium) exists for $A_i$ when $I_{i(t+1)} = I_{it}$ for $i = 1, 2$. Thus, equilibria are the roots of the two simultaneous polynomial equations

$$0 = \begin{aligned} & \frac{\alpha_1 b_0}{N} I_1^3 + \left( \alpha_1 b_0 \frac{I_2}{N} - b_0 \hat{c}_1 - \frac{\alpha_1 b_0}{2} \right) I_1^2 + (b_0 \hat{c}_1 \frac{N}{2} - \\ & \frac{b_0 \alpha_1 I_2}{2} - b_0 c_{12} I_2 - \gamma) I_1 + b_0 c_{12} I_2 \frac{N}{2} \end{aligned} \tag{24}$$

and

$$0 = \begin{aligned} &\frac{\alpha_2 b_0}{N} I_2^3 + \left(\alpha_2 b_0 \frac{I_1}{N} - b_0 \hat{c}_2 - \frac{\alpha_2 b_0}{2}\right) I_2^2 + (b_0 \hat{c}_2 \frac{N}{2} - \\ &\frac{b_0 \alpha_2 I_1}{2} - b_0 c_{12} I_1 - \gamma) I_2 + b_0 c_{12} I_1 \frac{N}{2} \end{aligned} \tag{25}$$

## Stability

To analyze local stability, we calculate the Jacobian of the differential equations corresponding to Eqs 21 and 22, where $I_{1(t+1)} = f_1(I_{1t}, I_{2t})$ and $I_{2(t+1)} = f_2(I_{2t}, I_{2t})$ and evaluate the Jacobian at the equilibrium. Differentiating each function with respect to each variable $I_1$ and $I_2$,

$$\frac{\partial f_1}{\partial I_1} = 3 \frac{\alpha_1 b_0}{N} I_1^2 + 2 \left(\alpha_1 b_0 \frac{I_2}{N} - b_0 \hat{c}_1 - \alpha_1 b_0 \frac{N_1}{N}\right) I_1 + \left(b_0 \hat{c}_1 N_1 - b_0 \alpha_1 I_2 \frac{N_1}{N} - b_0 c_{12} I_2 + 1 - \gamma\right), \tag{26}$$

$$\frac{\partial f_1}{\partial I_2} = \frac{b_0 \alpha_1}{N} I_1^2 - b_0 \alpha_1 \frac{N_1}{N} I_1 - b_0 c_{12} I_1 + b_0 c_{12} N_1, \tag{27}$$

$$\frac{\partial f_2}{\partial I_1} = \frac{b_0 \alpha_2}{N} I_2^2 - b_0 \alpha_2 \frac{N_2}{N} I_2 - b_0 c_{12} I_2 + b_0 c_{12} N_2, \tag{28}$$

$$\frac{\partial f_2}{\partial I_2} = 3 \frac{\alpha_2 b_0}{N} I_2^2 + 2 \left(\alpha_2 b_0 \frac{I_1}{N} - b_0 \hat{c}_2 - \alpha_2 b_0 \frac{N_2}{N}\right) I_2 + \left(b_0 \hat{c}_2 N_2 - b_0 \alpha_2 I_1 \frac{N_2}{N} - b_0 c_{12} I_1 + 1 - \gamma\right). \tag{29}$$

## Computational results

### The 2-population model

We first examine the numerical iteration of the discrete time SIS recursions (Eqs 21 and 22) without time-delay, and set default values for all parameters. No empirical research has analyzed comparative utility lost or gained as a function of contacts. Thus, the parameter set has been chosen to demonstrate possible dynamic regimes, rather than to parameterize COVID-19 or other infectious diseases. These default parameters are: $N_1 = 5000$, $N_2 = 5000$, $I_0 = 1$, $\gamma_1 = 0.1$, $\gamma_2 = 0.1$, $\hat{c}_1 = 0.02$, $\hat{c}_2 = 0.01$, $c_{12} = 0.005$, $\alpha_1 = 1$, $\alpha_{21} = 20$, $\alpha_{22} = 40$, $b_0 = 0.009$.

By increasing the transmissibility parameter $b_0$ from the default value, we see a progression of dynamical regimes across critical thresholds from simple convergence to cyclic behavior in 2, 4, and 6-point cycles, chaos, and collapse (Figs 1, 2 and 3). By increasing the contact rate between the 2 populations, $c_{12}$, there is a progression from simple convergence to a 2-pt cycle, 6-pt cycle, chaos, back to a 6-pt cycle, and finally an asynchronous 8-pt and 2-pt cycle (Figs 4 and 5). Bifurcation diagrams in Figs 2 and 5 indicate sustained dynamic patterns for a continuous parameter change in either $b_0$ (Fig 2) or $c_{12}$ (Fig 5). When $c_{12} = 0$ and $c_t^*$ is only based on prevalence in the same age category, we see what would happen in a homogeneous population without age structure with the same parameters (Fig 6). Without this interdependent age-

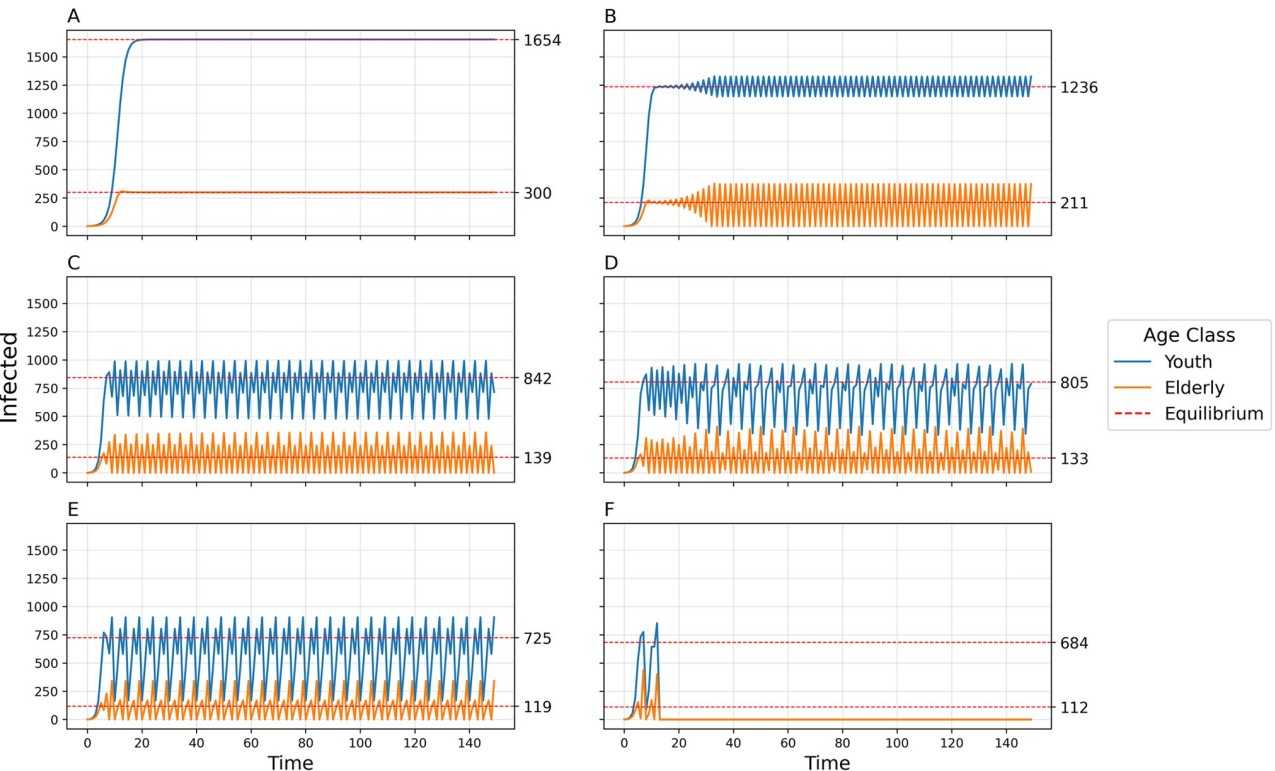

**Fig 1. Number of infected individuals over time for two-population age-structured model using default parameters and varying transmissibility $b_0$.** A) $b_0 = 0.009$, Convergence to stable equilibria; B) $b_0 = 0.013$, 2-point cycle; C) $b_0 = 0.02$, 4-point cycle; D) $b_0 = 0.021$, chaos; E) $b_0 = 0.0235$, 6-pt cycle; F) $b_0 = 0.025$, Collapse.

structure, Youth dynamics are even more complex, while Elderly dynamics have less complexity.

The aversion to infection for the elderly and the youth given by $\alpha_{22}$ and $\alpha_{21}$, respectively, may produce more complex dynamics at higher values, but the relationship between the two values is also important (Fig 3). Simpler dynamics (e.g., equilibrium and 2-point cycles) can be found closer to the line $y = x$ (i.e., $\alpha_{22} = \alpha_{21}$), while highly differentiated aversions to infection are more likely to produce chaos or oscillations of higher periodicity.

## The 3-population model

For our 3-age-class model, $A_1$ represents the youth, $A_2$ represents the middle-aged, and $A_3$ represents the elderly. Then the analogous equations to Eqs 6–13, for $k = 3$, are

$$M_t = \begin{bmatrix} c_{t1}^* & c_{12} & c_{13} \\ c_{12} & c_{t2}^* & c_{23} \\ c_{13} & c_{23} & c_{t3}^* \end{bmatrix}, \tag{30}$$

$$I_{1(t+1)} = I_{1t} + b_0 c_{1t}^* S_{1t} I_{1t} + b_0 c_{12} S_{1t} I_{2t} + b_0 c_{13} S_{1t} I_{3t} - \gamma I_{1t}, \tag{31}$$

$$I_{2(t+1)} = I_{2t} + b_0 c_{2t}^* S_{2t} I_{2t} + b_0 c_{12} S_{2t} I_{1t} + b_0 c_{23} S_{2t} I_{3t} - \gamma I_{2t}, \tag{32}$$

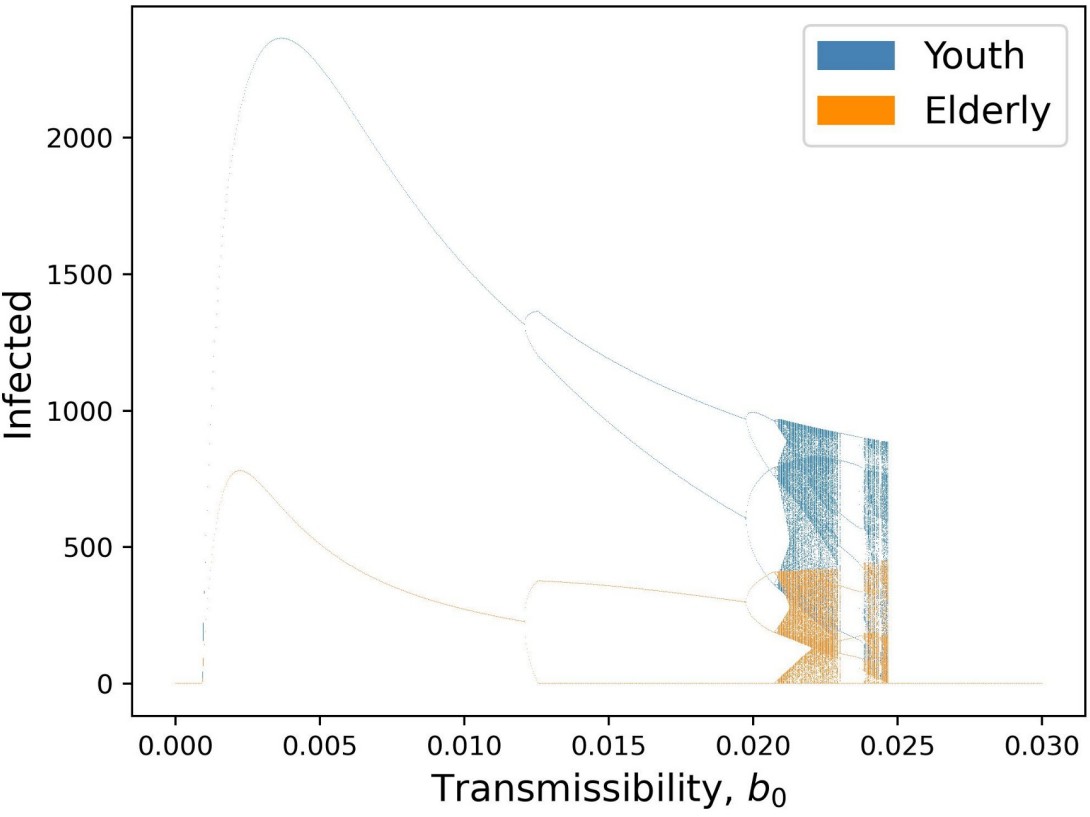

**Fig 2. Bifurcation diagram of equilibria, oscillatory dynamics, and chaotic behavior as a function of transmissibility $b_0$.**
Points in the figure represent sustained long-term results in the model; thus any $b_0$ value with, for example, two corresponding y-axis points represents a sustained oscillation between two values for infected individuals.

$$I_{3(t+1)} = I_{3t} + b_0 c_{3t}^* S_{3t} I_{3t} + b_0 c_{13} S_{3t} I_{1t} + b_0 c_{23} S_{3t} I_{2t} - \gamma I_{3t}, \tag{33}$$

with

$$S_{1t} + I_{1t} = N_1, S_{2t} + I_{2t} = N_2, S_{3t} + I_{3t} = N_3. \tag{34}$$

Equilibria for the system defined by Eqs 31–33 solve $I_{i(t+1)} = I_{it}$ for $i = [1, 2, 3]$ and are the roots of three simultaneous polynomial equations.

We set default values for parameters and initial conditions, such that $N_1 = 5000$, $N_2 = 5000$, $N_3 = 5000$, $I_0 = 1$, $\gamma_1 = 0.1$, $\gamma_2 = 0.1$, $\gamma_3 = 0.1$, $\hat{c}_1 = 0.02$, $\hat{c}_2 = 0.015$, $\hat{c}_3 = 0.01$, $c_{12} = 0.005$, $c_{13} = 0.003$, $c_{23} = 0.007$, $\alpha_1 = 1$, $\alpha_{21} = 20$, $\alpha_{22} = 30$, $\alpha_{23} = 40$, $b_0 = 0.01$.

By increasing the transmissibility $b_0$, the model goes from simple convergence to a 2-point cycle to chaos into a 2-pt cycle to collapse (Fig 7). By uniformly increasing the between-group contact rates $c_{12}$, $c_{13}$, and $c_{23}$, the model exhibits convergence, a 2-pt cycle, a 4-pt cycle, chaos, a 5-pt cycle, and a 6-pt cycle (Fig 8). The amplitude of oscillations and the variance of chaotic dynamics are greatest for the elderly population and least for the youth population. When a time-delay is introduced ($\Delta = 1$), dynamic complexity appears at smaller levels of $b_0$ (see Fig 9) than when $\Delta = 0$ (Fig 7).

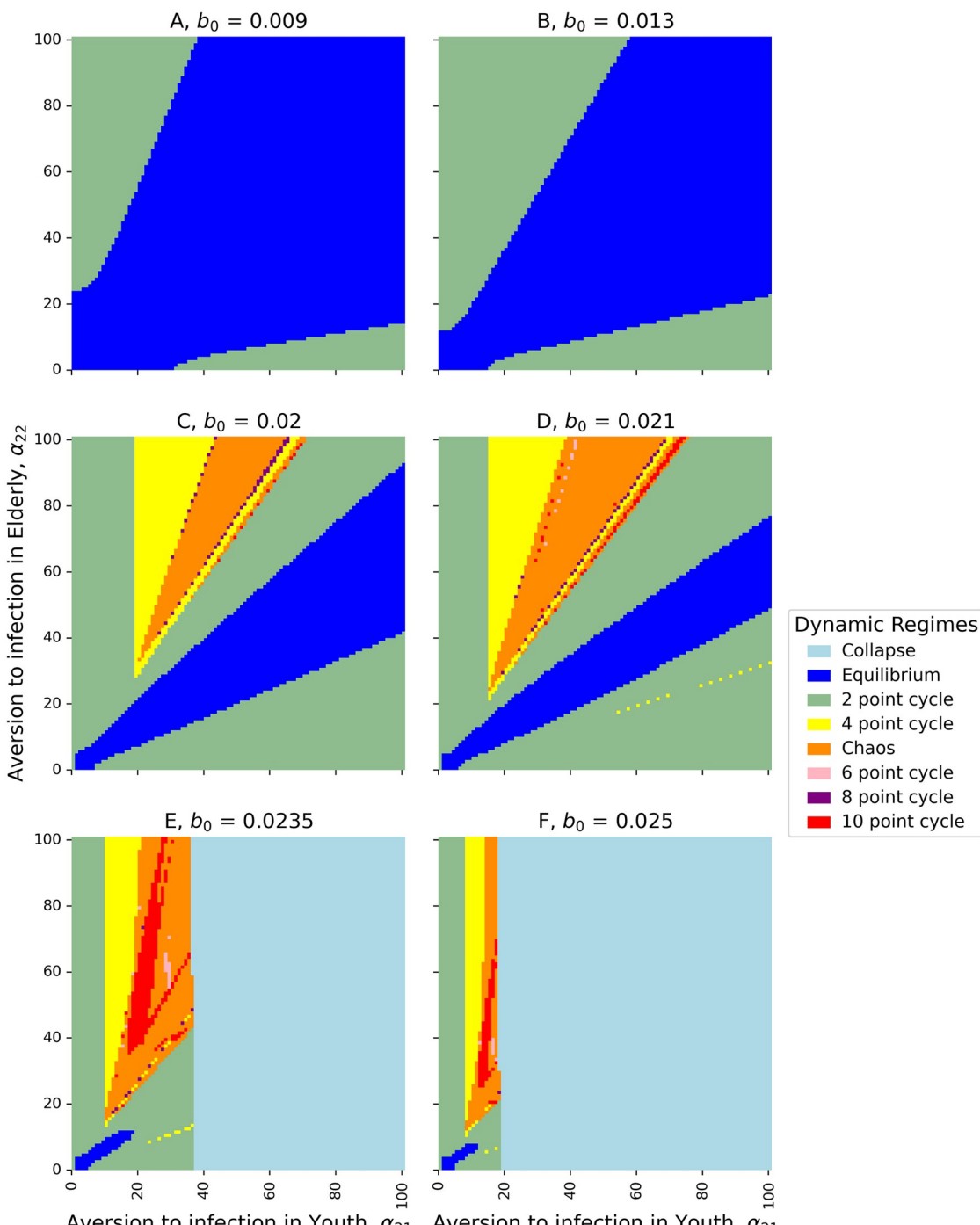

**Fig 3. Phase plane of non-zero equilibrium and stability.** Heat maps of dynamic regimes as a function of the relationship between aversion to infection in the youth ($\alpha_{21}$) and in the elderly ($\alpha_{22}$). Dynamic regimes are indicated by color and A–F are increasing in transmissibility ($b_0$): A) $b_0 = 0.009$; B) $b_0 = 0.013$; C) $b_0 = 0.02$; D) $b_0 = 0.021$; E) $b_0 = 0.0235$; F) $b_0 = 0.025$.

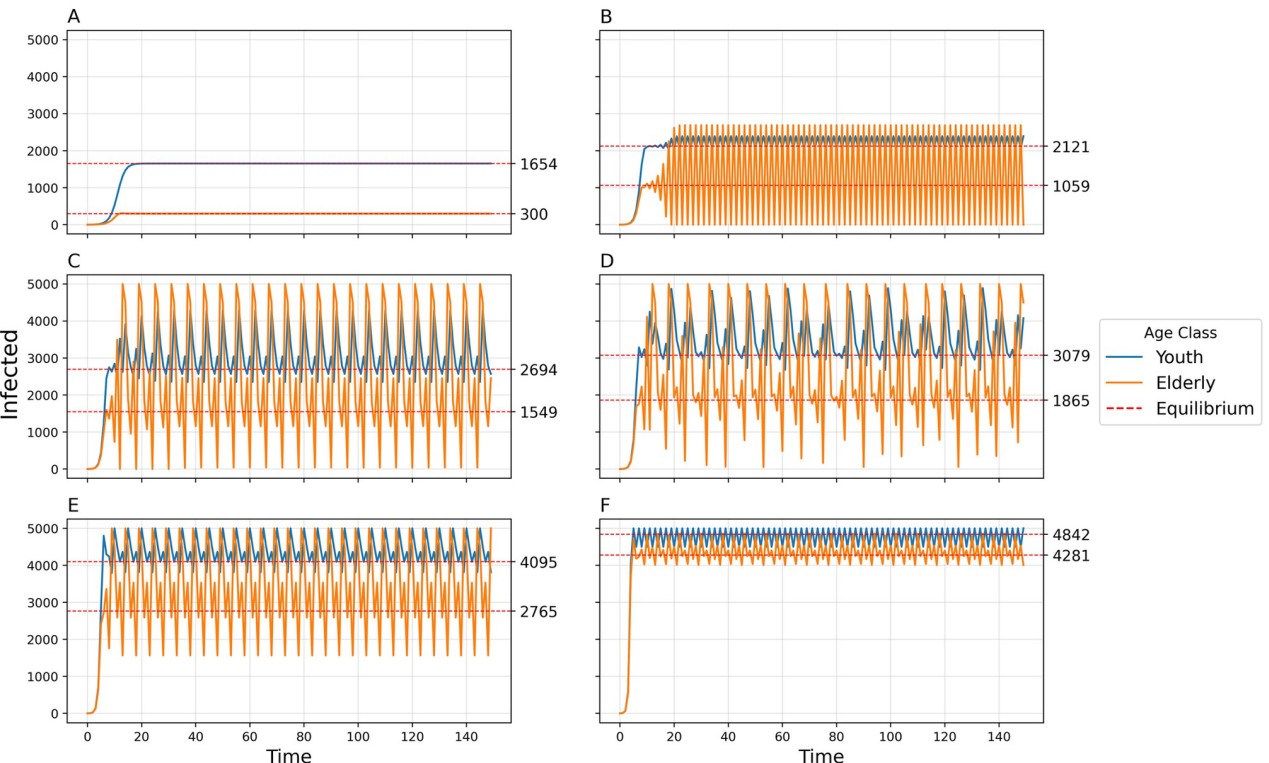

**Fig 4. Number of infected individuals over time for two-population age-structured model using default parameters and varying between-group contact rate $c_{12}$.** A) $c_{12} = 0.005$, Convergence to stable equilibria; B) $c_{12} = 0.025$, 2-point cycle; C) $c_{12} = 0.04$, 6-point cycle; D) $c_{12} = 0.05$, chaos; E) $c_{12} = 0.08$, 6-pt cycle; F) $c_{12} = 0.15$, 8-pt cycle for Old and 2-pt cycle for Young.

## Discussion

Our model has generalized previous work on adaptive behavior during epidemics to include age structure with heterogeneous aversion to infection and target contact rates. Results from the 2- and 3-population adaptive behavior models indicate that, for certain parameter values, stable equilibria can be reached for each sub-population $i$, including 0 for all age groups and an equilibrium between 0 and $N_i$. In the 2-population case, the equilibrium for each population can be derived numerically. With increasing values of $b_0$ and $c_{ij}$ (the transmissibility and between-group contact rate, respectively), the system may converge to a non-zero equilibrium, oscillate perpetually with 2-, 4-, 6-, 8-, and 10-period intervals, become chaotic, and collapse. Complexity of dynamics can be found at lower levels of transmissibility with greater differentiation between aversions to infection (Fig 3). For both the 2-population and 3-population models, the younger population has a higher non-zero equilibrium size than the older population, and the older population has greater amplitude of oscillations and variance of chaotic dynamics than the younger population.

Our model is built on a number of simplifying assumptions. First, the model is of the SIS type and does not consider an Exposed class (E) or a Recovered class (R). The inclusion of E or R (in, e.g., SEIRS models) may decrease the number of susceptible individuals and thus slow some of the oscillatory or chaotic dynamics found here. The inclusion of R for SIR models depletes susceptibles and prevents sustained dynamic patterns. Risk tolerance and reactions to shifting prevalence are stratified by age class, but assumed homogeneous within each class. In

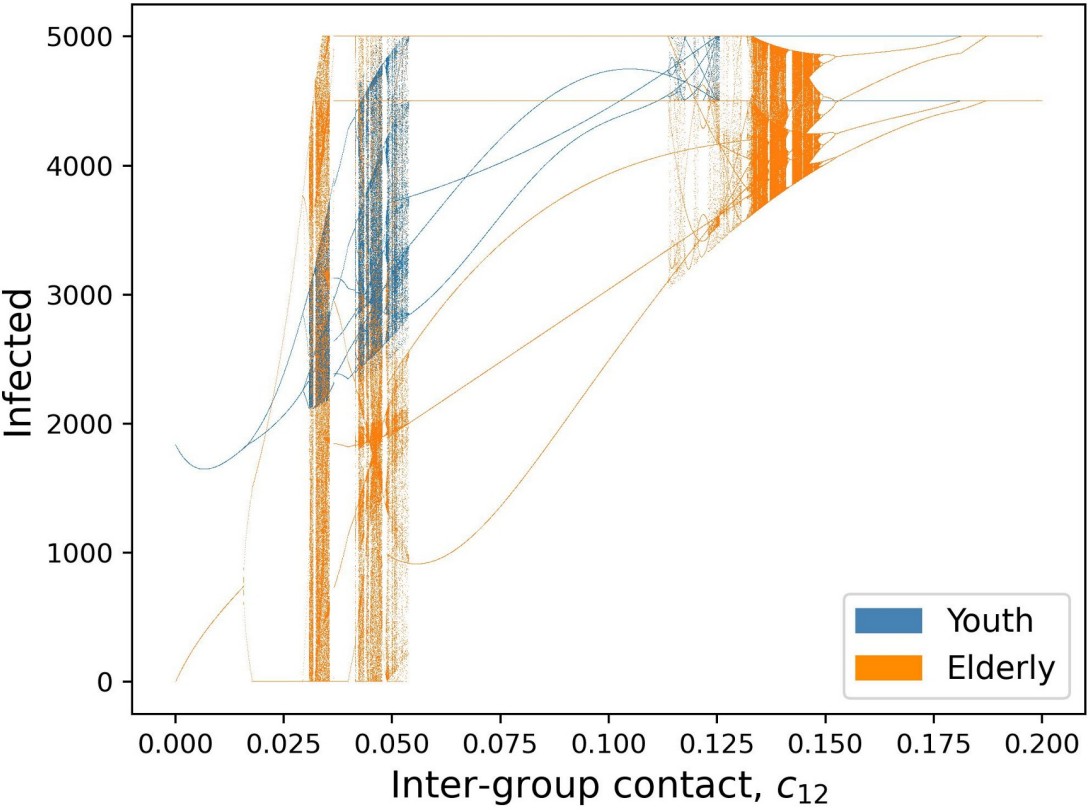

**Fig 5. Bifurcation diagram of equilibria, oscillatory dynamics, and chaotic behavior as a function of inter-group contact rate $c_{12}$.** Points in the figure represent sustained long-term results in the model; thus any $c_{12}$ value with, for example, two corresponding y-axis points represents a sustained oscillation between two values for infected individuals.

reality, individuals would have heterogeneous risk tolerance according to their age, political affiliation, and other characteristics. We modeled adaptive behavior with dynamic within-group contact rates, but assumed between-group contact rates were fixed. While this was justified by COVID-19 public health policies that controlled within-group contact more than between-group contact, between-group contact is also responsive to prevalence. Expanding this model to finer age categories than the ones used in this paper should include empirically-derived contact rates and differentiated optimization for between-group contacts. Further, we constrained the model with four mathematical assumptions (Eqs 10–13), some of which may not always hold in real-world scenarios. The relaxation of the above assumptions may yield different model outcomes. We note the model was not trained on real-world data to produce parameters as empirical data on adaptive behavior and incentives is lacking. While the dynamical regimes found here exist in theory, we do not know where in practice real-world disease systems may be. However, COVID-19 exhibited oscillatory patterns and chaotic dynamics [28], suggesting that this theoretical work may be directly relevant to diseases with strong adaptive behavior components.

Without contact with other groups, the Elderly age class exhibits simpler dynamics (e.g., monotonic convergence) (Fig 6). While the older population has a higher risk associated with infection, the younger population's lower aversion and higher baseline contact rates affect the epidemic dynamics in the older population. Thus, transmission in the young may not only

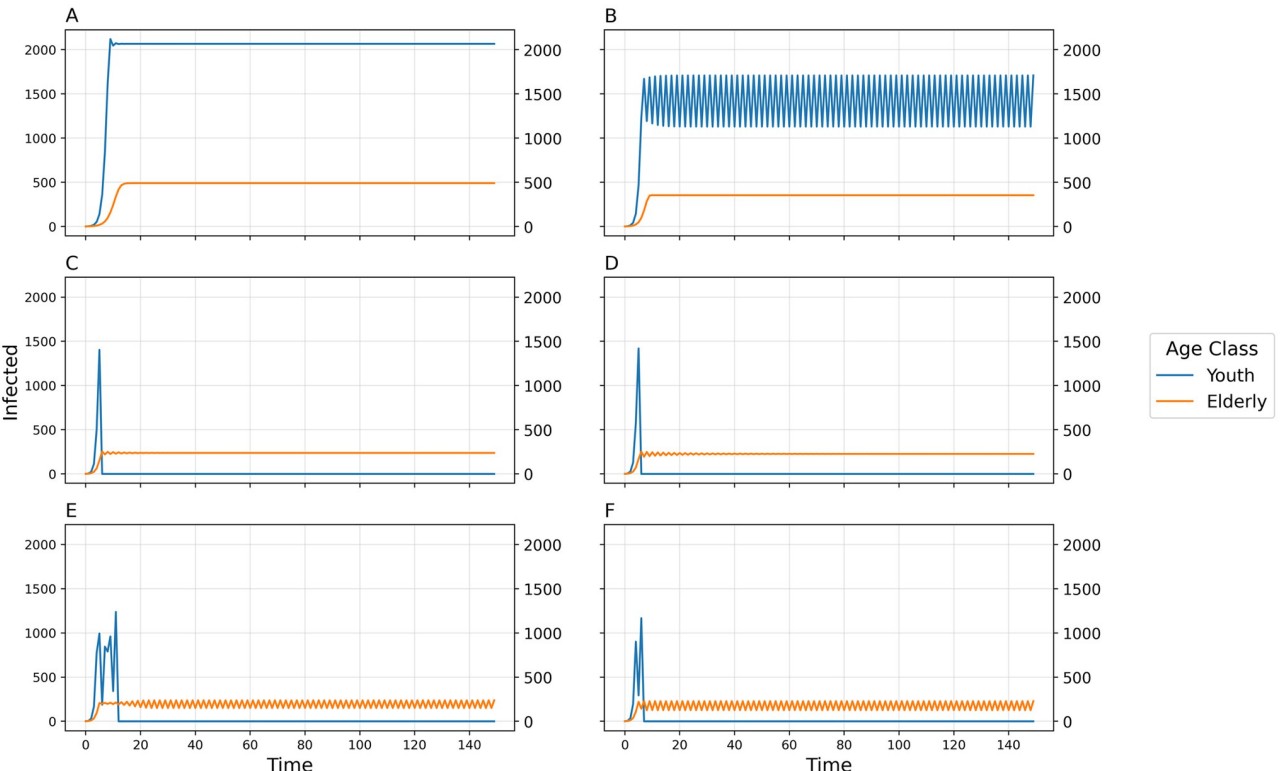

**Fig 6. Number of infected individuals for Youth and Elderly populations over time with default parameters, each population size set to N, no between-group contact (i.e., $c_{12} = 0$), and varying transmissibility $b_0$.** A) $b_0 = 0.009$; B) $b_0 = 0.013$ =,; C) $b_0 = 0.02$; D) $b_0 = 0.021$; E) $b_0 = 0.0235$; F) $b_0 = 0.025$.

lead to transmission in the elderly, but also increase the variance of the elderly dynamics and destabilize them. Indeed, even at lower levels of infection aversion for the elderly, the more differentiated the aversions to infection between the two age classes, the more complex the dynamics (Fig 3). Children are known to be the primary drivers of Influenza transmission, although severe morbidity and mortality are mostly seen in older age groups [39]. The importance of the youth in the dynamics of our model provides theoretical support for COVID-19 lockdown policies that reduce between-group interaction (i.e., $c_{ij}$), as found by Acemoglu et al. [15]. Transmission within the household is a key point of risk for the elderly and middle-aged, and high levels of transmission in the youth are likely to significantly affect disease outcomes in other age categories.

Our model may be usefully compared to other adaptive behavior models that look at sub-populations with differentiated characteristics or reactions to epidemic dynamics. It is worth noting that the justification for the bifurcated structure need not be restricted to age-based differentiation, but can also include political party affiliation, income levels, or other demographic, social, behavioral, physical, or geographic differences. Some studies use structured 2-population models with varying homophily and outgroup aversion or varying awareness separation and mixing separation [40, 41]. Results from these models indicate that heterogeneous populations, even when simply structured compartmentally as two populations, can produce greater complexity in epidemic dynamics, including large second

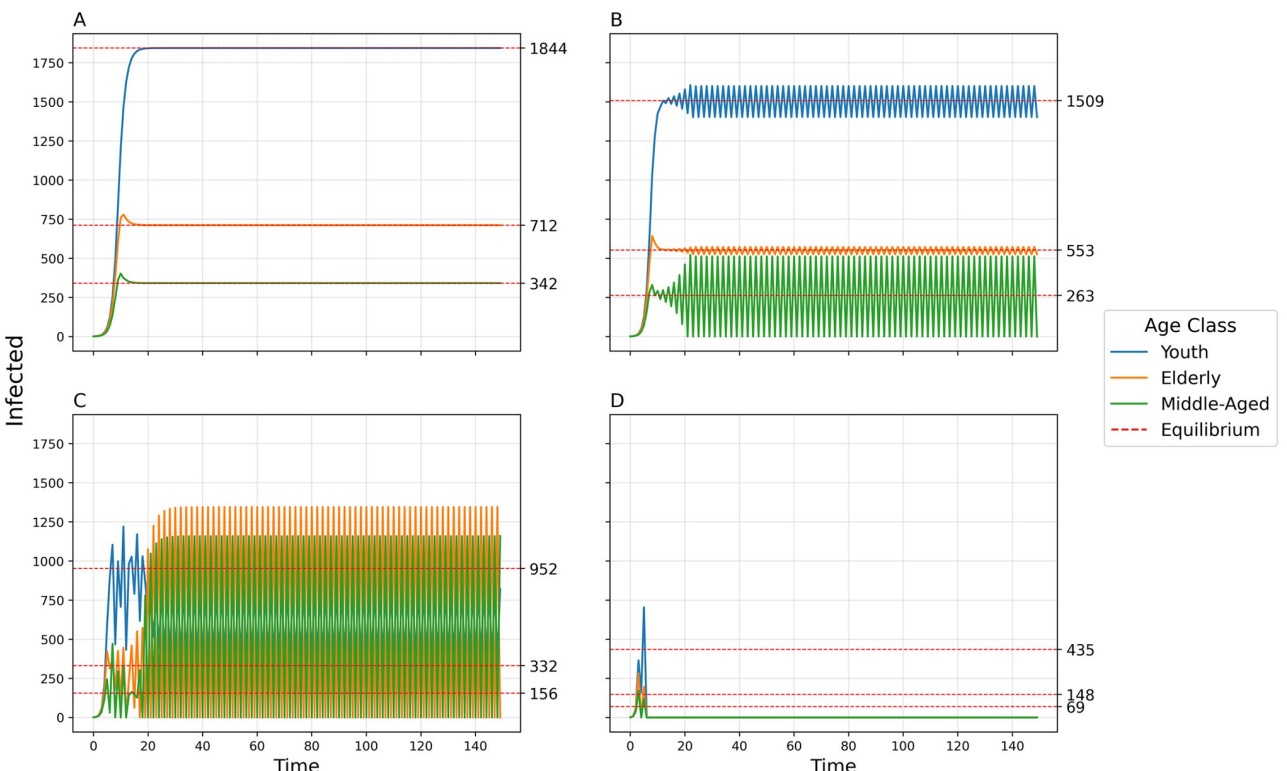

**Fig 7. Number of infected individuals over time in each of three populations using default parameters and varying transmissibility $b_0$.** A) $b_0$ = 0.01, Convergence to stable equilibria; B) $b_0$ = 0.013, 2-point cycle; C)$b_0$ = 0.022, Chaos into a 2-pt cycle cycle; D) $b_0$ = 0.05, Collapse.

waves and interconnected dual epidemics. Our results broadly agree with this theme: a 2-population model with varying infection aversion can produce more complexity at lower levels of transmissibility, and the difference between aversions can also drive complexity (e.g., Fig 3).

In our previous work [27], we compared the complex dynamics associated with endogenous behavior change during epidemics to similar theoretical behavior in ecological systems. For example, the logistic map is an extensively-studied simple model of population growth with limited resources that exhibits similar dynamics to our adaptive behavior models: convergence, oscillations, chaos, and collapse [42]. The driver of these dynamics in the logistic map is r, the growth rate. In our model, two of the principal drivers are $b_0$, the transmissibility, and $c_{ij}$, the between-group contact rate, as the epidemic's reproduction number $R_0$ is directly proportional to transmissiblity and contact rate. Often, when a single population is modeled in ecology (e.g., in a fishery [43]), the carrying capacity operates as an attractor, above which the population is attracted downwards and below which the population is attracted upwards. With endogenous behavior, the equilibrium, or indifference point, is also an attractor, below which the population is motivated to relax protective behaviors and above which the population is motivated to adopt protective behaviors. It may be productive to extend this parallel further in the case of multiple populations. For example, the predator-prey model considers two interdependent populations. When the prey population is high, the predator population grows, but when the prey are low in number, the predators decrease. While our epidemiological model does not include such direct

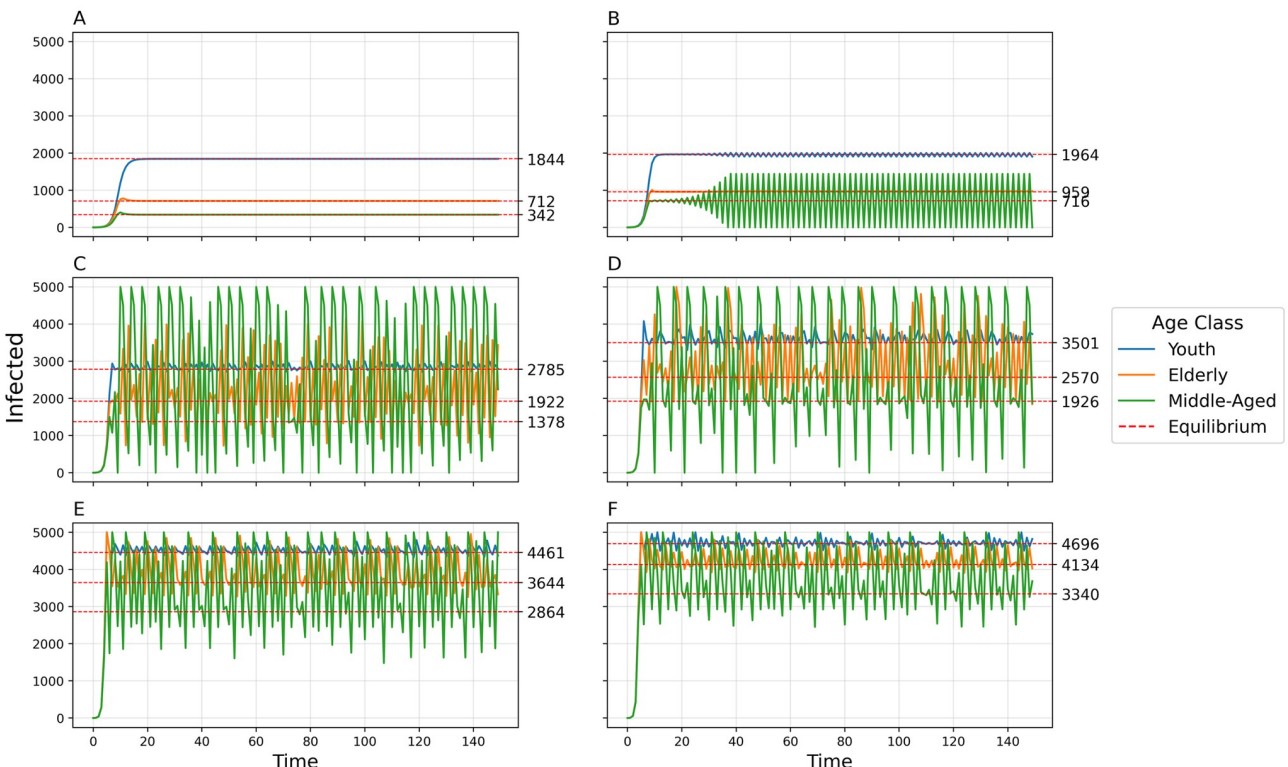

**Fig 8. Number of infected individuals over time in each of three populations using default parameters and varying between-group contact rates**
$c_{12}$, $c_{13}$, **and** $c_{23}$. A) $c_{12} = 0.005$, $c_{13} = 0.003$, $c_{23} = 0.007$, Convergence to stable equilibria; B) $c_{12} = 0.008$, $c_{13} = 0.01$, $c_{23} = 0.01$, 2-point cycle; C) $c_{12} = 0.022$, $c_{13} = 0.02$, $c_{23} = 0.024$, 4-point cycle; D) $c_{12} = 0.032$, $c_{13} = 0.03$, $c_{23} = 0.034$, Chaos; E) $c_{12} = 0.05$, $c_{13} = 0.048$, $c_{23} = 0.052$, 5-pt cycle; F) $c_{12} = 0.06$, $c_{13} = 0.058$, $c_{23} = 0.062$, 6-pt cycle.

competition or predation, the behavior is somewhat similar: when the young population has high prevalence of disease, the prevalence in the older population increases until the indifference point is crossed. When the young population has low prevalence, the prevalence in the older population follows suit. If a comparison between ecology and epidemiology is appropriate, it follows that careful study of the literature in theoretical ecology may provide insights relevant to epidemiology, a field with as yet comparatively little exploration of such complex system dynamics.

We recommend further theoretical work in both adaptive behavior modeling and complexity in epidemiology. A systems perspective may better represent the inherent complexities and heterogeneities of real-world epidemics. Social drivers of disease have been relatively neglected in the literature [2], but played an important role in COVID-19 outcomes. For example, the divergence of risk assessment by age class that characterizes our model was a social phenomenon, though biologically motivated. Other complex phenomena important to COVID-19 include simultaneous asynchronous epidemics, political bifurcation of attitudes and practices, and the co-evolution of the human immune system and the virus. As public health policies depend on our ability to forecast different scenarios under a high degree of uncertainty and complexity, such modeling will play an important role in improving policy and health outcomes in future epidemics.

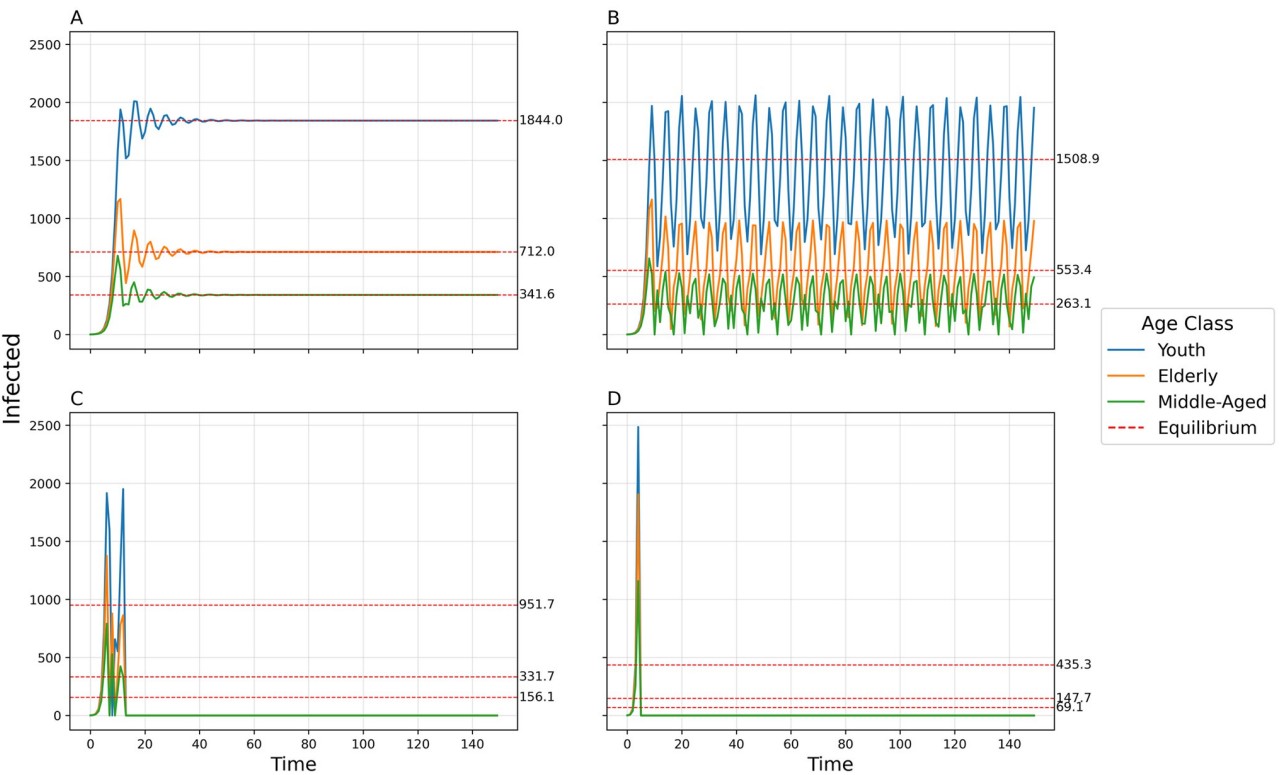

**Fig 9. Disease dynamics for the 3-population model when delta = 1.** Number of infected individuals over time in each of three populations using default parameters, time-delay $\Delta = 1$, and varying transmissibility $b_0$. A) $b_0 = 0.01$, Damped oscillator; B) $b_0 = 0.013$, Chaos; C) $b_0 = 0.022$, Collapse; D) $b_0 = 0.05$, Collapse.

## Author Contributions

**Conceptualization:** Ronan F. Arthur, Marcus W. Feldman.

**Data curation:** Ronan F. Arthur, Marcus W. Feldman.

**Formal analysis:** Ronan F. Arthur, May Levin, Alexandre Labrogere, Marcus W. Feldman.

**Funding acquisition:** Marcus W. Feldman.

**Investigation:** Ronan F. Arthur, May Levin, Alexandre Labrogere, Marcus W. Feldman.

**Methodology:** Ronan F. Arthur, May Levin, Alexandre Labrogere, Marcus W. Feldman.

**Project administration:** Ronan F. Arthur, Marcus W. Feldman.

**Resources:** Ronan F. Arthur.

**Software:** Ronan F. Arthur, May Levin, Alexandre Labrogere, Marcus W. Feldman.

**Supervision:** Ronan F. Arthur, Marcus W. Feldman.

**Validation:** Ronan F. Arthur, Marcus W. Feldman.

**Visualization:** Ronan F. Arthur, May Levin, Marcus W. Feldman.

**Writing – original draft:** Ronan F. Arthur, Marcus W. Feldman.

**Writing – review & editing:** Ronan F. Arthur, Marcus W. Feldman.

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
