## [Decision Letter · Decision Letter 0]

26 Jun 2023

Dear Dr. Feldman,

Thank you very much for submitting your manuscript "Age classes stratified by risk and adaptive behavior during epidemics" for consideration at PLOS Computational Biology.

As with all papers reviewed by the journal, your manuscript was reviewed by members of the editorial board and by several independent reviewers. In light of the reviews (below this email), we would like to invite the resubmission of a significantly-revised version that takes into account the reviewers' comments.

All reviewers found the work interesting but also raised a few concerns that need to be addressed. These include the robustness check of the analysis, a discussion of other disease models beyond the SIS model, and an examination of aggregate dynamic behavior across all age groups. We look forward to receiving your constructively revised manuscript.

We cannot make any decision about publication until we have seen the revised manuscript and your response to the reviewers' comments. Your revised manuscript is also likely to be sent to reviewers for further evaluation.

Sincerely,

Feng Fu

Academic Editor

PLOS Computational Biology

Natalia Komarova

Section Editor

PLOS Computational Biology

All reviewers found the work interesting but also raised a few concerns that need to be addressed. These include the robustness check of the analysis, a discussion of other disease models beyond the SIS model, and an examination of aggregate dynamic behavior across all age groups. We look forward to receiving your constructively revised manuscript.

Reviewer's Responses to Questions

**Comments to the Authors:**

Reviewer #1: The manuscript discusses how adaptive behavior can cause the emergence of chaos in an SIS model. The resuts are interesting and the manuscript is well-written, but I believe some further analysis is necessary for publication. In particular, for the utility function (equation (2.4)) a robustness analysis would be interesting. What happens if the cost function has a different form? Additionally, the Figures 2 and 4 need thicker lines to be legible. For Figures 1,3,5 and 7, the text needs to be larger.

Reviewer #2: This is a relatively novel article that explores the impact of heterogeneity in exposure patterns, mortality, and transmissibility among different age groups on epidemic outcomes. By changing the propagation rate and the contact parameters between age groups, it is well demonstrated that the adaptive behavior may cause the system to converge to non-zero equilibrium, and the behavior of period-doubling bifurcation, chaos and collapse may occur with the increase of parameters. I would like to recommend the paper for publication once the following problems are well addressed.

1. The classification of the three age groups is a little vague, and it is best to have a specific age range for the elderly, which needs to be explained.

2. The incubation state E is not considered, which may not be infectious. Whether it is necessary to add E state to the model needs to be explained separately.

3. The recovery rate from the infected category to the susceptible is the same for all age classes, namely, for all i, j. Whether it is a little unrealistic, the recovery time of older people should be longer than that of younger people.

4. Figure 1. Why does the higher the transmission rate b, the more it tends to collapse? It is better to explain it in the paper.

5. The explanation in Figure 5 is short of what needs to be explained. Are there any meaningful findings?

6. Why with the increase of b, there will be a collapse phenomenon, the article needs to analyze the reason for this phenomenon.

7. In the figure, infected should be explained. Is it a daily infection? Although it is reflected in the figure, it should be explained to distinguish it from cumulative infection. In addition, it should be explained why infected people oscillate around a certain value. Why is it stable around some value?

8. The innovation of the article needs to be highlighted, what is the difference from the previous work, or what is the extension of the work of others.

9. In Chapter 4, why are the parameters set this way in the numerical simulation, and what needs to be introduced.

10. Where in lines 186-187 does the influence of young people on old people's transmission need to be explained.

11. Since there is oscillating behavior between different age groups, is there also this period-doubling bifurcation behavior for the whole population N? You can also make a graph to illustrate that it is best to compare the oscillating behavior between the whole population of age group 2 and age group 3 with basically the same parameters.

Reviewer #3: The authors place attention on the effects of social and behavioral factors in driving the dynamics of infectious diseases, primarily examining contact mixing and severity-based aversion to infection. It is nice to see these matters incorporated into modeling, especially as willingness to engage in risk-mitigating behaviors was a highly contentious matter while attempting to control COVID-19 spread in the United States.

Specific comments:

1. Lines 94-97: I understand the need to avoid optimizing all elements of a contact matrix, but does this impose any limitations on the model, such as needing to define groups such that mixing will not change substantially during an epidemic? This is reasonable for the age groups selected by the authors (children, working-age adults, elderly adults), but would be more difficult to justify if using finer age groups, such as 10-year age bands.

2. Lines 134-135: How were these parameters selected? It would be beneficial if these values were chosen based on realistic resemblance to existing infectious diseases, whether through the parameter values themselves or by the disease dynamics produced through this model parameterization.

3. Similarly, it would be good to see if the results shown in Figures 1-6 are robust to other sets of baseline parameters. As the authors adjusted single or pairs of parameters, the others remained at these default values. Had different default values been chosen, would there be notable differences in the results?

4. This work presents how an SIS model is affected by social behaviors. However, it is common for infectious pathogens to confer at least temporary immunity following infection. While I recognize that extending all of the methods/results for an SIRS model could be unnecessarily complex and for little gain, it would be worthwhile in the discussion to describe how other model setups beyond an SIS model would likely be affected by the social and behavioral factors from this study.

**Have the authors made all data and (if applicable) computational code underlying the findings in their manuscript fully available?**

Reviewer #1: None

Reviewer #2: None

Reviewer #3: **No: **

PLOS authors have the option to publish the peer review history of their article (what does this mean?). If published, this will include your full peer review and any attached files.

Reviewer #1: No

Reviewer #2: No

Reviewer #3: No
---

## [Decision Letter · Decision Letter 1]

11 Aug 2023

Dear Dr. Feldman,

We are pleased to inform you that your manuscript 'Age-differentiated incentives for adaptive behavior during epidemics produce oscillatory and chaotic dynamics' has been provisionally accepted for publication in PLOS Computational Biology.

Please see some minor comments of one of the reviewers (including a missing reference), and apply as you see fit.

Best regards,

Feng Fu

Academic Editor

PLOS Computational Biology

Natalia Komarova

Section Editor

PLOS Computational Biology

Thank you for your great effort in revisions and all reviewers now unanimously recommend acceptance. Please note that R#2 has recommended some minor revisions (i.e., typos and text clarity) be taken care of in the finalized version sent to the production office.

Reviewer's Responses to Questions

**Comments to the Authors:**

Reviewer #1: My concerns have been addressed. I recommend the paper for publication.

Reviewer #2: The authors improved the paper and answered all of my questions. Thus, now I suggest publication.

Reviewer #3: Thank you for the revised manuscript. My previous comments have been adequately addressed. Some minor suggestions for further clarification:

1. In the abstract and in the last sentence of the Introduction section (lines 79-81), it would be helpful to include a specific definition of what adaptive behaviors are being examined in this study. As written, it relies on referring to previous work, but is crucial in setting up the primary purpose of the paper.

2. The manuscript presents and SIS model and does note how adding other compartments such as E or R would likely impact the results, but primarily references the behaviors shown in the model to diseases that often use one or both of those compartments in modeling, notably measles, influenza, and COVID-19. If it is possible to relate these results to diseases that are more commonly fit to SIS models, it would strengthen the actual applicability of these concepts.

3. In the Introduction section, two of the references do not seem to appear correctly, appearing as a "?" in the in-text citations.

**Have the authors made all data and (if applicable) computational code underlying the findings in their manuscript fully available?**

Reviewer #1: None

Reviewer #2: None

Reviewer #3: Yes

PLOS authors have the option to publish the peer review history of their article (what does this mean?). If published, this will include your full peer review and any attached files.

Reviewer #1: No

Reviewer #2: No

Reviewer #3: No

---

## [Editor Report · Acceptance letter]

22 Aug 2023

PCOMPBIOL-D-23-00840R1 

Age-differentiated incentives for adaptive behavior during epidemics produce oscillatory and chaotic dynamics

Dear Dr Feldman,

I am pleased to inform you that your manuscript has been formally accepted for publication in PLOS Computational Biology. Your manuscript is now with our production department and you will be notified of the publication date in due course.

With kind regards,

Zsofi Zombor
